# Arrestins: A Small Family of Multi-Functional Proteins

**DOI:** 10.3390/ijms25116284

**Published:** 2024-06-06

**Authors:** Vsevolod V. Gurevich

**Affiliations:** Department of Pharmacology, Vanderbilt University, Nashville, TN 37232, USA; vsevolod.gurevich@vanderbilt.edu

**Keywords:** arrestin, GPCR, MAP kinase, microtubules, ubiquitin ligase

## Abstract

The first member of the arrestin family, visual arrestin-1, was discovered in the late 1970s. Later, the other three mammalian subtypes were identified and cloned. The first described function was regulation of G protein-coupled receptor (GPCR) signaling: arrestins bind active phosphorylated GPCRs, blocking their coupling to G proteins. It was later discovered that receptor-bound and free arrestins interact with numerous proteins, regulating GPCR trafficking and various signaling pathways, including those that determine cell fate. Arrestins have no enzymatic activity; they function by organizing multi-protein complexes and localizing their interaction partners to particular cellular compartments. Today we understand the molecular mechanism of arrestin interactions with GPCRs better than the mechanisms underlying other functions. However, even limited knowledge enabled the construction of signaling-biased arrestin mutants and extraction of biologically active monofunctional peptides from these multifunctional proteins. Manipulation of cellular signaling with arrestin-based tools has research and likely therapeutic potential: re-engineered proteins and their parts can produce effects that conventional small-molecule drugs cannot.

## 1. Introduction—Discovery of Arrestin Subtypes

Mammals express four arrestin subtypes [1] that were discovered as proteins that selectively bind activated phosphorylated forms of cognate G protein-coupled receptors (GPCRs) [2]. The first member of the arrestin family, like all other key players in GPCR signaling, was discovered in rod photoreceptors [3,4]. The reason for this is simple: rod photoreceptors express all proteins of the visual signaling cascade at orders-of-magnitude higher concentrations than the levels of their non-visual homologs in “normal” cells [5]. Arrestin-1 was discovered before its biological function was established: it was first described and purified as soluble antigen (or S-antigen) targeted by antibodies in autoimmune disease uveitis [6,7]. This is why its gene is called *Sag* (stands for *soluble antigen*) in the HUGO database. In a different line of research, pioneering studies by Kuhn identified a 48 kDa protein (he called it just that), which specifically binds rhodopsin activated by light [3]. His lab then showed that the binding of 48 kDa protein increases upon rhodopsin phosphorylation [4] and that this protein blocks the activation of photoreceptor phosphodiesterase by light-activated rhodopsin [8]. Later, it was established that S-antigen and 48 kDa protein are one and the same [9,10]. Because S-antigen/48 kDa protein stopped (“arrested”) rhodopsin signaling, a new name, “arrestin”, was proposed [11], which is now widely accepted. Early on, visual arrestin was reported to bind ATP [12] and Ca^2+^ [13], but subsequent experiments did not confirm these claims [14]. Within a few years, visual arrestin was cloned [15,16]. Its linear sequence suggested possible structural homology with the α-subunit of transducin [15,17], but the crystal structures of these proteins [18,19,20,21] disproved this hypothesis.

Rod photoreceptors consist of two parts: the outer segment, where rhodopsin and proteins of the signaling cascade are localized, and the rest of the cell, connected to the outer segment via a narrow cilium [22]. In the 1980s, it was discovered that in the dark, most of arrestin-1 is outside of the outer segment, whereas most of transducin, a visual G protein that couples to rhodopsin, stays in it [23,24,25]. Light induces transducin translocation away from the outer segment and concomitant arrestin movement into this compartment. Biologically, this makes perfect sense: both the removal of a signal transducer and the arrival of a quencher reduce rod sensitivity to light when it becomes bright. Although active transport was hypothesized to be involved, both arrestin [26] and transducin [27] were shown to move by diffusion, in an energy-independent manner (reviewed in [28]). The calculation of the enormous amounts of ATP that would have been required to move huge numbers of these protein molecules in rods by active transport clearly shows that diffusion is the only viable option [29]. This does not exclude possible active gating at the connecting cilium, although demonstrated ATP independence makes it unlikely [26,27].

The primary structure of the hamster β2-adrenergic receptor (β2AR) [30] revealed its significant homology with the light receptor rhodopsin and the presence of seven membrane-spanning regions [31]. This structural homology suggested that both proteins belong to the same family (now called rhodopsin-like GPCRs, or class A [32]). This finding gave rise to the idea, which is now accepted in the field, that the key signaling and desensitization mechanisms first described for rhodopsin are shared by class A GPCRs. The kinase that phosphorylates agonist-activated β2AR, a phenomenon resembling activation-dependent phosphorylation of rhodopsin by the rhodopsin kinase [33], was cloned in 1989 [34]. By analogy with the rhodopsin kinase (which was cloned later [35]; its modern systematic name is GRK1), this kinase was named βARK (β2AR kinase) [34] (its modern systematic name is GRK2). Purified βARK was shown to reduce β2AR-mediated activation of Gs only marginally. This finding suggested that there must be a functional equivalent of visual arrestin that binds phosphorylated β2AR [36]. The protein that fulfils this role was cloned by homology with visual arrestin [37]. It demonstrated clear preference for β2AR over rhodopsin, and therefore was termed β-arrestin; to discriminate between the two, the original arrestin from photoreceptor cells was renamed visual or rod arrestin [37,38]. These arrestin names turned out to be misleading. When the arrestin from cone photoreceptors (termed cone [39] or X-arrestin [40]) was cloned, the term “visual” became equivocal. The finding that the expression of cone-specific arrestin in cone photoreceptors was relatively low [41], whereas cones contain a ~50-fold higher level of the presumably rod-specific subtype [42], demonstrated that the terms “cone” and “rod” arrestin are inaccurate. Another non-visual arrestin subtype, which was called β-arrestin2, was cloned later [43] (the same protein was independently described as hTHY-ARRX [44]), and the first non-visual arrestin subtype was retroactively renamed β-arrestin1. Experimental demonstration that both non-visual arrestins bind not only β2AR, but numerous other GPCRs [45,46,47], showed that the term “β-arrestin” is also inaccurate. Arrestin nomenclature used in the literature is confusing. Below I use systematic names of arrestin proteins, proposed in 1994 [48]: the four mammalian arrestin subtypes are numbered in the order of their cloning: arrestin-1 (previously used names include S-antigen, 48 kDa protein, visual or rod arrestin), arrestin-2 (β-arrestin or β-arrestin1), arrestin-3 (β-arrestin2 or hTHY-ARRX), and arrestin-4 (cone or X-arrestin). Systematic names of arrestin subtypes contain only numbers indicating the order of cloning, which is not ambiguous, and they do not imply anything that can be proved incorrect. Below, I focus on the structural basis of arrestin functions elucidated by my laboratory and many others.

## 2. Arrestin Structure and Modes of Self-Association

### 2.1. Structure of Free Arrestins

The first crystal structure of arrestin was published in 1998 (PDB ID 1AYR; [21]). It correctly revealed the overall fold, but contained one mistake: the authors placed the arrestin-1 N-terminus where in later, higher-resolution 1999 structure (PDB ID 1CF1; [20]) its C-terminus was shown to be located. Subsequently solved structures of arrestin-2 (PDB ID 1G4M; [49] and 1JSY; [50]), arrestin-4 (PDB ID 1SUJ; [51]), arrestin-3 (PDB ID 3P2D; [52]), squid arrestin that binds unphosphorylated squid rhodopsin (PDB ID 6BK9; [53]), and the more recent structure of bovine arrestin-1, where a greater part of the C-terminus than in all other structures was resolved (PDB ID 7JSM; [54]), demonstrated that the fold is conserved in the family. All arrestins are elongated proteins with the long axis of 70–75 A (Figure 1). The molecule consists of two domains (usually termed N- and C-domain), and the C-terminus that folds back, making contact with the N-domain. Each domain is a bent ‘sandwich’ of seven β-strands, with concave and convex sides, and loops of varying lengths connecting these β-strands (Figure 1). Arrestins have only one short α-helix attached to the N-domain. Interestingly, a closely structurally related arrestin-domain containing protein, retromer subunit Vps26, lacks this α-helix (and some other structural features) [55].

The structures showed that the orientation of the two domains relative to each other in the basal state of the arrestin molecule is maintained by two “clasps”. One is the “polar core” (the term from [20]): five residues with interacting charged side chains (two positive and three negative charges) in the interface between the two domains (Figure 1). These charges are virtually solvent-excluded [20], which is an unusual arrangement for a soluble protein. The other is a “three-element interaction” (this term is also from [20]) of the β-strand I plus the α-helix from the N-domain, and the C-terminus. This interaction is mediated by large hydrophobic side chains in all three elements (Figure 1). Interestingly, the interaction between arrestin-1 N- and C-termini was predicted based on mutagenesis years before the structure was solved [56]. The two domains are connected by a 12-residue hinge region, the length of which is conserved in the family (Figure 1) [1]. The first few N-terminal residues and two parts of the C-terminus (proximal, between the C-domain and the three-element interaction, and distal, downstream of the three-element interaction) are never resolved in structures. Apparently, these parts of the molecule are highly flexible and do not have a preferred conformation.

### 2.2. Self-Association of Arrestins

Three out of four mammalian subtypes [1], arrestin-1, -2, and -3, self-associate [57]. The oligomers formed by these are dramatically different. Arrestin-1 from different species forms dimers and tetramers (dimers of dimers) [58,59], with the solution tetramer being a closed rhomboid [60,61]. Bovine, mouse, and human arrestin-1 oligomerize similarly, although dimerization and tetramerization constants differ between species [59]. Its self-association is inhibited by an abundant cytoplasmic metabolite, inositol-hexakisphosphate (IP_6_), whereas the same IP_6_ facilitates oligomerization of both non-visual subtypes [58]. Arrestin-2 forms long chains with no apparent limit [62], likely resembling those observed in IP_6_-soaked crystals of arrestin-2 [63]. In both cases, protomers apparently retain the basal conformation observed in crystal structures [61,62]. In contrast, arrestin-3 forms trimers, where each protomer is in receptor-bound-like conformation [64].

Considering that each subtype forms a unique type of oligomer, it appears likely that oligomeric arrestins must have special functions. However, the data on the function of arrestin oligomers are spotty. While only monomeric arrestin-1 binds rhodopsin, both monomers and oligomers bind microtubules [60]. Considering that the identified tubulin-binding surface of arrestin-1 largely overlaps with that mediating rhodopsin binding (see Section 3.1) [65], which is shielded by sister protomers in the tetramer [61], this suggests that oligomeric arrestin-1 uses alternative elements (that remain to be elucidated) for the interaction with polymerized tubulin. As rod photoreceptors express an enormous amount of arrestin-1 (intracellular concentration up to 2 mM) [66], the hypothesis that the oligomers are the storage form [67] reducing the monomer concentration appears plausible. Indeed, an oligomerization-deficient arrestin-1 mutant causes concentration-dependent (but light-independent) death of rods [68] at concentrations where oligomerizing wild-type protein does not harm these cells [66].

Monomeric arrestin-2 and -3 can enter the nucleus [69], whereas oligomers, including arrestin-2–arrestin-3 heterodimers, were reported to remain in the cytoplasm [63,70]. Inhibition of arrestin-3 oligomerization by mutating IP_6_ binding sites was reported to reduce its ability to interact with ubiquitin ligase Mdm2 and its p53-dependent anti-proliferative action without significantly affecting the interactions with GPCRs [71]. However, inhibition of arrestin-3 self-association by a different set of mutations was reported to reduce the binding to β2AR and ERK1/2 [72]. Finally, oligomers of arrestin-3 were reported to impair the clearance of pathological tau and increase tau aggregation [73].

Possible other specific functions of the oligomers of arrestin-1, -2, and -3 that monomeric forms of these proteins cannot fulfil still need to be elucidated. Multi-functionality of relatively small (~45 kDa) arrestin proteins [74], along with the number of their interaction partners that exceeds 100 in the case of arrestin-2 and -3 [75], suggests that there likely are proteins in the cell that preferentially interact with oligomerized arrestins. This hypothesis needs to be tested experimentally.

## 3. GPCR Binding

The first discovered function of arrestins was selective binding to active phosphorylated GPCRs, which blocked receptor interactions with G proteins by simple competition [76,77]. This aspect was confirmed by the very first crystal structures: Gs in the complex with β2AR [78] and arrestin-1 in the complex with rhodopsin [79] engage the same cavity that opens on the cytoplasmic side between the helices upon GPCR activation [80]. Among known functions of arrestin proteins, their interaction with receptors is the most studied.

### 3.1. Receptor-Binding Elements

The identification of arrestin-1 residues involved in rhodopsin binding started before the structure was solved. As arrestin preferentially binds phosphorylated receptor, and phosphates are negatively charged, the first studies using differential acetylation [81] and site-directed mutagenesis [82] focused on positively charged arrestin residues. Acetylation identified 14 lysines in arrestin-1 that were protected by phosphorhodopsin binding; these lysines are scattered throughout the linear sequence from position 2 to 367 [81]. The role of many of these lysines in rhodopsin binding was later confirmed by mutagenesis [83]. Mutagenesis study focused on the nine positively charged residues within 161–190 region that was previously implicated in phosphate binding [84]. Lysines 163, 166, and 167 were identified in both studies [81,82]. Lys-157 in arrestin-2 (homologous to Lys-163 in arrestin-1) at the distal end of the N-domain (part of the N-edge, Figure 1) was recently identified as one of the residues that comes close to the parathyroid hormone receptor PTH1R using in-cell cross-linking of the two proteins via reactive unnatural amino acids [85].

Arrestin-1 binds rhodopsin much better than the muscarinic M2 receptor, whereas arrestin-2 demonstrates the opposite preference [47]. On the assumption that elements affecting receptor selectivity must be directly involved in the interaction, a series of arrestin-1/2 chimeras were constructed, and their binding to rhodopsin, β2AR, and M2 muscarinic receptors was tested [46,47]. The results suggested that the middle portion of both arrestins plays an important role in receptor selection, whereas 47 (in arrestin-1) or 43 (in arrestin-2) N-terminal, as well as 59 (arrestin-1) and 78 (arrestin-2) C-terminal residues do not [46], consistent with previous findings that the N- and C-terminal parts of arrestin-1 are regulatory elements [56,84,86]. However, the design of this study did not allow the role of the elements conserved in both subtypes to be identified.

Mapping receptor-binding elements on the structures solved later suggests that the concave sides of the two arrestin domains and loops in the central crest on the same side of the molecule (Figure 1) constitute the receptor-binding surface. This side of arrestin-1 and arrestin-2 faces bound GPCRs in solved structures of the complexes [79,87,88,89,90,91,92,93,94,95].

### 3.2. Arrestin Selectivity for the Active Phosphorylated GPCRs

Arrestins preferentially bind active phosphorylated GPCRs, demonstrating much lower binding to other functional forms of the same receptor (reviewed in [96,97]) (Figure 2). The molecular mechanism underlying this selectivity was extensively studied, mostly in arrestin-1 that demonstrates the highest binding selectivity for the preferred form of rhodopsin (Figure 2).

#### 3.2.1. “Coincidence Detector” Mechanism

Properly folded functional arrestins can be expressed in cell-free translation, where proteins with high specific activity can be generated by replacing a cold amino acid with a hot one. The use of radioactive arrestins in direct binding assay with purified rhodopsin [56,82,84,86,98] or other GPCRs [46,47,99,100] ensures femtomolar sensitivity of the assay, much greater than other methods used to measure the interaction [101,102,103]. In the case of arrestin-1 binding to rhodopsin, this assay revealed that although light-activated phosphorylated rhodopsin (P-Rh*) is the preferred partner, there is detectable binding to inactive (dark) phosphorhodopsin (P-Rh) and light-activated unphosphorylated rhodopsin (Rh*), whereas in the case of inactive unphosphorylated rhodopsin, negligible binding was detected [84,86] (Figure 2). This suggested that arrestin-1 must have independent sites recognizing the activation state of rhodopsin and receptor-attached phosphates, which mediate its binding to Rh* and inactive P-Rh, respectively. However, arrestin-1 binding to P-Rh* exceeds that to P-Rh or Rh* 10–20-fold (Figure 2). This differential is too large to be explained by two-site cooperativity. Hence, the model of sequential multi-site binding was proposed: phosphate and active receptor binding sites were suggested to serve as sensors. Low-affinity binding to P-Rh or Rh*, both of which can engage only one sensor, results in rapid dissociation. P-Rh* engages both sensors simultaneously, which triggers arrestin-1 transition into high-affinity receptor-binding conformation, bringing additional elements into contact with rhodopsin, thereby increasing the energy of the interaction, yielding high-affinity binding [84]. This model (reviewed in [96]) satisfactorily explained arrestin preference and therefore was widely accepted in the field.

Based on the structure of a GPCR that simultaneously interacts with arrestin and G protein [104], where the arrestin engages exclusively phosphorylated receptor C-terminus, it was hypothesized that arrestins first bind the elements containing receptor-attached phosphates (“tail only” configuration in [104]) and then bind the transmembrane cavity of the receptor. Several lines of evidence contradict this idea. First, arrestin-1 binds Rh*, and arrestin-2 and -3 bind active unphosphorylated non-visual GPCRs (none of which has any attached phosphates) a lot better than inactive forms of the same receptors (Figure 2) [46,47,82,84,86,99,100,105,106]. Second, a variety of phosphorylation-independent mutants of arrestin-1, -2, and -3 bind cognate unphosphorylated receptors with high affinity [46,82,84,86,98,99,100,106,107,108,109,110,111,112,113,114]. Third, several vertebrate wild-type GPCRs bind wild-type non-visual arrestins without receptor phosphorylation [115,116,117,118], as do invertebrate rhodopsins [53,119,120,121]. In all of these cases, arrestins must bind unphosphorylated receptor elements first and last, as there are no receptor-attached phosphates to assist in the process. The original hypothesis that arrestins can bind either phosphates on the receptor or the active GPCR first, engaging the other feature second [84], does not contradict available data.

#### 3.2.2. Identification of Sensors in Arrestins

Phosphate and active receptor sensors needed to be identified. The finger loop of arrestin-1 (Figure 1) that binds in the cavity between transmembrane helices [79], which opens upon the activation of rhodopsin [80] and other GPCRs [122], was shown to serve as the sensor of receptor activation [109]. All three classes of proteins that preferentially bind activated GPCRs engage this cavity: G proteins [78], GRKs [123], and arrestins [79], as the appearance of this cavity is a hallmark of GPCR activation. In solved structures of arrestin-1 and -2 complexes with GPCRs the finger loop can remain unstructured (with M2 muscarinic [89], β_1_-adrenergic [91], V2 vasopressin [92], and CB1 cannabinoid [95] receptors) or form a short α-helix (with rhodopsin [79], neurotensin [88,90], and serotonin 5HT_2B_ receptors). Non-visual arrestin-2 and -3 bind many different GPCRs (reviewed in [124,125]). It appears that this loop can engage this cavity in different receptors in a distinct manner, as mutations in it differentially affect the interaction: e.g., in arrestin-3, Asp70Pro kills the binding to M2 muscarinic, but only reduces the binding to dopamine D2 receptors, whereas Leu69Arg totally eliminates the binding to M2, but does not appreciably affect the binding to D2 [64]. Finger loops of both arrestin-2 and -3 have residues with diverse side chains (large hydrophobic, hydrophilic uncharged, as well as positively and negatively charged [109]). This, along with the conformational flexibility of the finger loop, likely allows it to fit into the cavity in different receptors lined with side chains of different chemical nature.

The identification of the arrestin phosphate sensor was convoluted. The early finding that charge neutralization of Arg175 in arrestin-1, which is located within a cluster of phosphate-binding, positively-charged residues, yields a mutant that does not require phosphates (binds Rh* fairly well) [82] focused the attention on this residue. Its replacement with the other 19 residues showed that with the positive charge in this position, arrestin-1 requires rhodopsin phosphorylation for tight binding; with the other residues it shows varying levels of phosphorylation independence; and with the negative charge in this position, the binding becomes independent of the presence of rhodopsin-attached phosphates [98]. This suggested that it likely interacts with a negatively charged residue in the basal state of arrestin, and that phosphate breaks this interaction by neutralizing its charge [126], like the mutations, thereby allowing arrestin to bind with high affinity. Charge reversal of homologous arginines in arrestin-2 [100,114,127] and -3 [99,128] also made these subtypes phosphorylation-independent, suggesting that a similar mechanism operates in all arrestins. The finding that this arginine is located in the polar core of arrestin-1 [20,21], and its homologs are in the polar cores of the other subtypes [49,50,51,52] (Figure 1), supported the idea of its functional significance. The structures identified potential negatively charged counterparts of this arginine. Follow-up mutagenesis showed that Asp296 is the key in the bovine arrestin-1: its charge reversal also yielded phosphorylation-independent mutants, whereas simultaneous reversal of both charges, which restored the salt bridge in an opposite configuration, yielded arrestin-1 that required rhodopsin phosphorylation, essentially like the wild type [111]. Arginine substitution of Asp296, as well as its homologs in mouse arrestin-1 and non-visual subtypes, greatly increased the binding to phosphorylated and unphosphorylated receptors [113,128]. All of these data were consistent with the idea that the polar core salt bridge between Arg175 and Asp296 (and their homologs in the other arrestins) serves as the phosphate sensor: receptor-attached phosphates break this bridge, enabling arrestin transition into high-affinity binding state (this idea was first suggested in [111]; reviewed in detail in [96]). This model looked attractive because it made perfect sense biologically: the sequences of the cytoplasmic elements of GPCRs vary greatly [32], while attached phosphates are the only common theme, so that this mechanism explains how the non-visual subtypes can bind hundreds of different GPCRs.

While in all structures of the arrestin complexes with receptors [79,87,88,89,90,91,92,93,95,104], phosphorylated receptor peptides [126,129], and with IP_6_ [64] the polar core is broken, in none of them the polar core arginine interacts with a phosphate. These data buried that model. However, a residue in the lariat loop (which supplies two negative charges to the polar core), Lys294 in arrestin-2 (Figure 1), and homologous Lys301 in mouse arrestin-1 were found in contact with the phosphate in the bound receptor (rhodopsin [87], neurotensin receptor [90], or engineered multi-phosphorylated V2 receptor peptide attached to the M2 muscarinic [89] and β_2_-adrenergic receptor [104]) or one of the phosphates in IP_6_ [64] or bound phosphopeptide [129]. This suggested the idea that the phosphate of the activator can destabilize the polar core in a different way: by pulling out the lariat loop, thereby removing two negative charges from their basal positions. However, the charge of this lysine (Lys295 and Lys300 in bovine arrestin-3 and -1, respectively) was eliminated (Lys→Ala) and reversed (Lys→Glu), with no detectable negative effect on the receptor binding [113,128]. These experimental results buried yet another beautiful idea.

All arrestins have two lysines in the β-strand I, right next to the hydrophobic residues mediating its interactions with the α-helix and the C-terminus (three-element interaction in Figure 1) [1]. Alanine substitutions of these lysines were shown to prevent the binding of wild-type arrestin-1 to P-Rh*, while having minimal effect on the binding of several structurally distinct phosphorylation-independent mutants [106]. At least one of these lysines was found to contact a receptor-attached phosphate in all structures where the phosphorylated receptor elements were resolved. Lysines in homologous positions are conserved in all arrestins, from *C. elegans* to mammals [130]. Collectively, this evidence makes β-strand I lysines the most likely candidates for the role of the phosphate sensor in arrestins [113]. Plausible mechanism of their action was proposed in 2000 [106]: even a small shift of β-strand I due to the pull on these lysines (or either one of them) by a receptor-attached phosphate would move the hydrophobic side chains holding the α-helix and the C-terminus out of position favorable for these interactions, destabilizing the three-element interaction (Figure 1), thereby facilitating the release of the C-terminus, which removes another conserved arginine (Arg382 in bovine arrestin-1 and its homologs in other arrestins) from the polar core, thereby destabilizing it, as well. Mutagenesis [99,108,110,113,128] shows that the release of the arrestin C-terminus is sufficient to facilitate arrestin binding to phosphorylated and unphosphorylated GPCRs.

Modeling suggests that both the polar core and three-element interaction need to be destabilized by the receptor to allow arrestin transition into receptor-bound conformation [131]. It appears that both of these are destabilized by the receptor-attached phosphates. However, recent mutagenesis studies suggest that there are additional mechanisms enhancing arrestin-1 selectivity for P-Rh*. Several arrestin-1 residues that are not part of either sensor specifically suppress its binding to Rh* [110,112]. Apparently, arrestin-1 selectivity for P-Rh* is so important biologically, that some native residues do this even at the expense of somewhat reducing the binding to the preferred target P-Rh* (i.e., their replacement increases P-Rh* binding [110,112]). As both arrestin-2 and -3 are a lot less selective for the active phosphorylated form of their cognate GPCRs, as compared to the inactive phosphorylated form (Figure 2), this additional mechanism might not exist in non-visual subtypes.

#### 3.2.3. The Role of Receptor-Attached Phosphates

In most cases, arrestins bind active phosphorylated mammalian GPCRs (reviewed in [96,124]). However, there are exceptions: arrestin binding to M1 muscarinic [117], substance P [115], or leukotriene B4 [116] receptors were reported not to require phosphorylation for arrestin binding. While alanine substitution of the phosphorylatable serines and threonines in β2AR reduces (but does not eliminate) arrestin-2 and -3 binding, elimination of phosphorylation sites in M2 muscarinic receptor does not appreciably change the binding of both non-visual arrestins [118]. The action of receptor-attached phosphates on the phosphate sensor of arrestins and their role in the interaction should be distinguished: the elimination of the two β-strand I lysines greatly reduces the binding of arrestin-1, -2, and -3 to P-Rh*, but does not appreciably affect the binding of the two non-visual subtypes to β2AR, M2 muscarinic, or D2 dopamine receptors [118]. Thus, in the case of β2AR, attached phosphates participate in the interaction not only via triggering the phosphate sensor, but likely via interaction with other phosphate-binding arrestin elements. Negatively charged phosphates at physiological pH must interact with positively charged arrestin residues (Lys, Arg, possibly His). Lys, Arg, and His can interact with negatively charged glutamic and aspartic acids, which are fairly abundant in the cytoplasmic elements of most GPCRs. Thus, Asp and Glu in the receptor can serve as substitutes for the GRK-attached phosphates. Indeed, negatively charged side chains were shown to participate in the arrestin binding of leukotriene B4 [116] and luteinizing hormone [132] receptors. Several Glu and Asp side chains of rhodopsin were found in contact with bound arrestin-1 in the complex [87].

In *Drosophila*, photoreceptor arrestins (two different subtypes are expressed) are as critical for signal termination as arrestin-1 is in vertebrates [133]. Yet the phosphorylation of fly rhodopsin is not required for arrestin binding, whereas its activation by light is [119,120,121]. This appears to be the general rule in invertebrates. The two lysines in the β-strand I that most likely serve as the phosphate sensor are conserved in invertebrate arrestins [130]. Charged residues of the polar core are less conserved, and the polar core in the only solved structure of an invertebrate arrestin is composed of an unconventional set of side chains [53]. Yet the three-element interaction and the polar core are present in the structure of squid arrestin [53], both of which likely need to be destabilized to allow high-affinity receptor binding. It is tempting to speculate that negatively charged residues in the invertebrate rhodopsins act on arrestins in lieu of receptor-attached phosphates. However, as fly arrestins selectively bind light-activated forms of rhodopsins, this hypothesis requires an additional assumption that arrestin-binding, negatively charged side chains in rhodopsin are inaccessible when it is inactive and become accessible upon light activation. To test whether this is the case, the structures of a fly rhodopsin in inactive and activated state are needed.

Both vertebrate non-visual arrestins preferentially bind phosphorylated forms of their cognate receptors (Figure 2). Receptor phosphorylation alone, without activation, greatly increases the binding of arrestin-2 and -3, in sharp contrast to exceptional selectivity of arrestin-1 for P-Rh* (Figure 2). This raises a question of how receptor-attached phosphates facilitate the binding of arrestin-2 and -3, considering that phosphate-binding lysines in the β-strand I [118] and the lariat loop [128] do not appear to be required. This is particularly intriguing considering that at least arrestin-2 can hold onto the receptor while interacting solely with its phosphorylated element (admittedly, with the help of bound antibodies that are not present in cells) [104]. This question can be addressed only by methods revealing the dynamics of the process, not by structures of the complexes that show the final result, but not the sequence of events whereby it is achieved. Importantly, GPCR activation without phosphorylation induces lower arrestin-2 and -3 binding than phosphorylation without activation (Figure 2), suggesting that activation-dependent exposure of negative charges in the receptor (as proposed above for invertebrate rhodopsins) is unlikely to play a significant role in the recruitment of vertebrate non-visual subtypes to GPCRs.

#### 3.2.4. Conformational Rearrangements Associated with Receptor Binding

The idea that arrestin-1 binding to rhodopsin involves a global conformational change was proposed more than 30 years ago on the basis of unusually high Arrhenius activation energy [103]. Soon thereafter, it was demonstrated that the binding to phosphorhodopsin involves the release of the arrestin-1 C-terminus [101,134]. The finding that C-terminal truncation greatly increases rhodopsin binding of arrestin-1 [84,86,108] suggested that the release of its C-terminus is necessary for the binding. Enhanced rhodopsin binding of the C-terminally truncated splice variant of arrestin-1, p44 [135,136], supported this idea. Enhanced binding of arrestin-2 and -3 with deleted C-termini to their cognate GPCRs [99,100,128] indicated that this part of the mechanism is conserved in the arrestin family. Double electron–electron resonance (DEER) (pulse electron paramagnetic resonance (EPR) technique) studies of arrestin-1 [137,138], -2, and -3 [139] revealed receptor binding-induced release of the C-termini of all three subtypes. Interestingly, while in the case of arrestin-1, the released C-terminus likely “flops around” (wide distributions of distances between the C-terminus and points in the rest of the molecule), in arrestin-2 and -3, the released C-terminus appears to have a preferred position. A recent single-molecule study of arrestin-2 suggests that there might be two distinct preferred positions of its released C-terminus [140].

The comparison of the structures of free arrestins [20,21,49,50,51,52,53,54] and GPCR-bound arrestin-1 [79,87] and -2 [88,89,90,91,92,93] reveals that the release of the arrestin C-terminus is not the only conformational change associated with GPCR binding. In a receptor-bound state, the two domains twist relative to each other by ~20°, as compared to their position in free arrestins. Similar domain twisting was observed in one of the structures of short arrestin-1 splice variant p44 [141] and in the IP_6_-induced trimer of arrestin-3 [64], suggesting that arrestin can achieve this conformation independently of GPCRs. Several additional conformational rearrangements in receptor-bound arrestins were revealed by EPR [142] and the structures of the arrestin-receptor complexes [79,87,88,89,90,91,92,93,95], the most prominent of which is a large movement of the middle loop (originally termed 139-loop in arrestin-1 [142,143]) (Figure 1) from its basal position overhanging the cavity of the C-domain in the direction of the N-domain.

When a protein engages the elements in both arrestin domains (as has been shown for many arrestin-binding partners [144,145,146]), it is easy to explain its conformational preference for receptor-bound or basal arrestins: only in one of these conformations, the position of interacting elements would be favorable for the binding. Based on this idea, one can predict that partners interacting with only one domain should not have a significant preference for either of these arrestin conformations. This needs to be tested experimentally. The difficulty is that no arrestin partner so far has been shown to interact with only one domain.

### 3.3. Receptor Preference of Arrestins

Vertebrates have two non-visual arrestin subtypes that apparently interact with hundreds of different GPCRs [1], whereas invertebrates and lower chordates express only one [130]. It is equally unclear why there are so few (as compared to a much greater variety of G proteins [147] and GRKs [148]) and why vertebrates retained two non-visual arrestin subtypes for millions of years of evolution, whereas invertebrates manage with one just fine.

The question what makes an arrestin ‘like’ particular receptors and ‘dislike’ others was first asked taking advantage of the fact that arrestin-1 binds rhodopsin well, but demonstrates relatively low binding to M2 muscarinic receptor, whereas arrestin-2 has the opposite preference [105]. The analysis of the binding of a series of arrestin-1/2 chimeras to these two receptors identified two elements on the concave side of the N- (~40 residues) and C-domain (~30 residues) that determine receptor preference (Figure 3). Swapping these elements between arrestin-1 and -2 completely reversed their receptor specificity [105]. Like the rest of the sequence, these regions are highly homologous in different arrestin subtypes [1]. Site-directed mutagenesis identified ten residues in these two elements that play key roles in the receptor preference [149]. Manipulation of these residues by mutagenesis greatly affected receptor specificity, enabling the construction of arrestin-3 variants that, in contrast to fairly promiscuous parental protein, demonstrate up to 60-fold preference for some GPCRs over others [150,151]. Several additional residues in the finger loop that binds the cytoplasmic cavity of active GPCRs also appear to serve the same purpose [109]. These data suggest the feasibility of narrowing receptor specificity of non-visual subtypes, constructing mutants that target few receptors, or possibly even individual GPCRs [152].

### 3.4. Structure of the Arrestin-GPCR Complexes: Possible Functional Implications

Due to high conservation of the sequence [1,130] and overall fold [20,21,49,50,51,52,53] of arrestin proteins, as well as the conservation of sequences [32] and the 3D structures of the core regions of GPCRs [32], the arrestin-GPCR complexes were expected to be similar. This turned out not to be the case. On the global level, the orientation of bound arrestin relative to the receptor core (seven transmembrane helices) was similar in case of rhodopsin [79,87], β_1_-adrenergic [91], engineered M2 muscarinic [89], and parathyroid hormone [85] receptors. However, in complex with neurotensin receptor NTSR1, bound arrestin-2 was turned by 85–90° relative to this orientation [88,90]. The orientation of arrestin-2 bound to serotonin 5HT_2B_ [93], vasopressin V2 [92], and cannabinoid CB1 [95] receptors was found to be intermediate between these two extremes. The tilt of the bound arrestin relative to the plane of the membrane where GPCRs are localized also varied in different structures. In addition, a number of subtler differences were detected. For example, the finger loop, which inserts into the cytoplasmic cavity between helices that opens upon receptor activation [80,122], was found to from a short α-helix in arrestins bound to rhodopsin [79], NTSR1 [88,90], and 5HT_2B_ [93] receptor, but not in complex with M2 [89], β_1_-adrenergic [91], V2 [92], and CB1 [95] receptors. It should be noted that the same non-visual subtype, arrestin-2, was in complexes with NTSR1, 5HT_2B_, M2, β_1_-adrenergic, V2, and CB1 receptors. In-cell crosslinking with reactive unnatural amino acids suggests that while different GPCRs, or even the same GPCR activated by different agonists, bind arrestins in a distinct manner, both non-visual arrestins bind the same receptor in a similar orientation [153].

The structures imply that there is only one shape of each arrestin-receptor complex. However, biophysical data suggest otherwise. DEER distance measurements between Y74 in the second transmembrane helix of rhodopsin (Ballesteros–Weinstein [154] numbering 2.41) and three points in different elements of bound arresin-1 molecule suggest that several “flavors” of the complex exist simultaneously: in all cases, multiple distances were detected [79]. The most populated distances matched the structure, indicating that it represents the prevalent variety of the complex, but the presence of the other distances indicates that alternative complexes, not resolved in structure, co-exist with it. Similarly, DEER measurements between the α-helix of bound arrestin-1 and several points in the rhodopsin C-terminus yielded multiple distances in each case [87]. Again, the most populated distances matched the structure [87], indicating yet again that the structure represents the prevalent shape of the complex, but it is not the only one existing.

Recently, arrestin-2 interactions with parathyroid hormone receptor PTHR1 were analyzed in the biologically relevant environment of a living cell using cross-linking between a reactive unnatural amino acid in numerous positions in arrestin-2 and cysteines in different positions on the cytoplasmic side of PTHR1 [85]. This study identified 136 intermolecular proximity points between the two proteins. No single conformation of the complex can bring all of them within cross-linking distance at the same time. Molecular dynamics simulations suggest that the complex is very flexible. All of the cross-linked residues come within the necessary distance in the sum total of possible configurations of the complex [85]. These data indicate that numerous shapes of the complex exist at any given moment in the cell.

Thus, the idea that the complex of each receptor–arrestin combination has a specific shape contradicts the evidence obtained both with purified proteins in vitro and in biologically relevant situations in cells. The most parsimonious explanation of the data is that the two proteins can form several alternative complexes, only one of which is resolved in the structure in each case. In fact, this idea is the basis for the “barcode hypothesis” [155,156], which is fairly popular in the field [131,157,158,159,160,161]. Arrestins need only three receptor-attached phosphates for tight binding [162,163], while most GPCRs have a lot more phosphorylatable serines and threonines on their cytoplasmic elements (see [87,164]). Thus, in the cell, multiple forms of each GPCR competent to bind arrestins can be generated. The barcode hypothesis posits that differentially phosphorylated GPCRs (likely by different GRKs [156,165,166]) form different complexes with arrestins with distinct functional capabilities [155,156,165,166]. Indeed, different patterns of phospho-mimetics in the D1 dopamine receptor appear to channel arrestin-mediated signaling to either ERK1/2 or Src [166]. Different positions of bound arrestin-2 and -3 relative to atypical chemokine receptor ACKR3 differentially phosphorylated by GRK2 or GRK5 were also documented [94].

Collectively, available data suggest that multiple structural variants of the complex of the same arrestin with the same receptor coexist in the cell, each likely mediating distinct functions. Structures of the same differentially phosphorylated receptor with the same arrestin are needed to test this attractive idea. So far, such structures were solved only in case of ACKR3 [94].

## 4. Arrestins in GPCR Trafficking

The role of both non-visual arrestins in GPCR internalization [167] and their interactions with clathrin, the key component of the coated pits [168], were discovered in 1996. Arrestins bind another major component of the internalization machinery, clathrin adaptor AP2 [169], although interactions with these two proteins appear to play differential roles in the process of receptor internalization [170]. These findings provided a mechanistic basis for the earlier suggested role of GRKs and arrestins in GPCR endocytosis ([171] and references therein). Clathrin- and AP2-binding sites are localized on the C-termini of both non-visual arrestins [168,169,170], which are released upon receptor binding, making these sites more accessible. This is biologically important, as separately expressed arrestin-2 C-terminus carrying these binding sites effectively competes with the GPCR-arrestin complexes, suppressing arrestin-mediated receptor internalization [172]. It is important to note that clathrin can interact with arrestins in basal conformation, as was demonstrated by the binding of purified non-visual arrestins with clathrin cages in the absence of GPCRs [168] and by receptor-independent arrestin–clathrin interactions necessary to disassemble focal adhesions [173].

Not all GPCRs that bind arrestins internalize with their help. For example, phosphorylated M2 muscarinic receptor robustly binds arrestin-2 [46,47,127,174]. This is important for its desensitization, but M2 internalizes via an arrestin-independent mechanism [175]. Yet it has been experimentally shown that M2 can also internalize via an arrestin-dependent pathway [176]. Thus, it appears that the M2 receptor itself carries an internalization signal that is stronger than that of the exposed C-terminus of bound arrestin, and that this signal wins the competition in the cell. This situation is not unique for the M2 receptor. Another GPCR, N-formyl peptide receptor, also robustly binds arrestins upon activation and phosphorylation [177,178], but internalizes in the absence of both non-visual arrestins (although, in this case, it does not recycle) [179].

Thus, non-visual arrestins are key players in the internalization of many, but not all, GPCRs. Arrestins appear to play a role in the return of internalized receptor to the plasma membrane, possibly via recruitment of deubiquitinating enzymes to the receptor [180,181], as GPCR deubiquitination is necessary for recycling.

## 5. Arrestin Interactions with Non-Receptor-Binding Partners

Arrestins bind numerous proteins involved in cell signaling and trafficking that are not GPCRs. Both non-visual subtypes were shown to bind >100 different proteins each [75]. However, in the field, the concept of the functional cycle of arrestins is still dominated by their binding to GPCRs. This is likely because the first described functional and structural states of arrestins were basal (free) and GPCR-bound [96,124,182]. The studies describing arrestin-mediated activation of protein kinases c-Src, JNK3, and ERK1/2 suggested that the signaling was triggered by GPCR activation, i.e., was mediated by the arrestin–receptor complex [183,184,185,186,187]. These pioneering studies established a new paradigm of the G protein-independent arrestin-mediated GPCR signaling [188]. However, the data were interpreted as indicating that free arrestins in their basal conformation do not do anything biologically important. Based on this assumption, the basal conformation of arrestins is routinely called “inactive”. It should be kept in mind that at least two binding partners, E3 ubiquitin ligases Mdm2 [69] and parkin [146], preferentially bind arrestins in their basal conformation. Moreover, the “footprints” of microtubules [65] and calmodulin [189] on arrestins largely overlap with the receptor-binding surface. Also, arrestin-3 mutants with enhanced receptor binding are either weak facilitators of JNK3 activation, or they completely lack the ability to facilitate the activation of JNK3 [128,190]. These findings suggest that free arrestins in their basal state are biologically active. Most reviews on the subject also implied that only non-visual arrestins have signaling functions in cells, whereas the only function of visual arrestin-1 and -4 is to bind active phosphorylated photopigments, terminating their signaling via cognate G proteins [22]. Experimental evidence that arrestin-1 and arrestin-4 bind Mdm2 and JNK3 [69,191], microtubules [65], and calmodulin [189]; that arrestin-1 binds NSF and enolase-1 and stimulates the activity of these enzymes in rod photoreceptors [192,193,194]; and that arrestin-1 binds AP2 in photoreceptors with high enough affinity to deplete it and cause rod death [195] suggest that visual subtypes are also multi-functional proteins. The structural basis and functional significance of the majority of arrestin interactions with non-GPCR partners remain unknown, so I focus on the interactions for which at least some data are available.

### 5.1. Microtubules

The binding of arrestin-1 to microtubules was described in 2005 [26]. Rod photoreceptors consist of a specialized outer segment where rhodopsin and the rest of the signaling cascade is localized, an inner segment rich in mitochondria and microtubules, a cell body containing the nucleus, and a synaptic terminal contacting the bipolar cells [22]. Rods express an enormous amount of arrestin-1 (the intracellular concentration reaches ~2 mM [66]), which exceeds the levels of non-visual arrestins in “normal” cells by ~10,000 times [196,197]. In the dark-adapted rod, the bulk of arrestin-1 is localized outside of the outer segment, with the highest concentration in the inner segment bound to microtubules [26]. Light activates rhodopsin, which is then phosphorylated by rhodopsin kinase [22], generating P-Rh*, the preferred target of arrestin-1 [29]. Illumination causes massive translocation of arrestin-1 to the outer segment due to P-Rh* binding [26]. The structural basis of arrestin-1 binding to tubulin was elucidated [198]. Comparable affinity of arrestin-1 for polymerized and unpolymerized tubulin (K_D_ 40 and 65 μM, respectively) suggested that arrestin-1 interacts with a single αβ-dimer of tubulin. The deletion of the C-terminus does not appreciably affect the binding, suggesting that this element is not involved in the interaction [198]. Mutagenesis and EPR data indicate that numerous residues on the concave side of both domains are involved in microtubule binding [198]. As this is the GPCR-binding surface (Figure 3), these data suggest that only free arrestin-1 can bind tubulin and explain why microtubule binding, similar to GPCRs, induces the release of the C-terminus [198]. The conformation of microtubule-bound arrestin differs from both basal and receptor-bound: deletions in the inter-domain hinge that suppress rhodopsin binding of arrestin-1 [199] tend to increase its binding to microtubules [198].

Subsequent studies revealed that all arrestin subtypes bind microtubules, with arrestin-3 demonstrating the highest binding [65]. Separated N- and C-domains of arrestin-2 and -3 also bound microtubules [65]. The K_D_ of arrestin-2 for microtubules and unpolymerized tubulin was found to be in the 26–28 and 35–50 μM range, respectively, similar to arrestin-1. As in the case of arrestin-1, increasing deletions in the inter-domain hinge of arrestin-2 and -3 progressively reduced their receptor binding, but did not negatively impact their binding to microtubules [65]. Several functional consequences of arrestin binding to microtubules were also reported. First, arrestins (except cone-specific arrestin-4) recruited ERK1/2 to the cytoskeleton, but did not recruit its upstream activating kinases Raf1 and MEK1, which resulted in significant decrease in ERK1/2 activation level [65]. Second, arrestin-1, -2, and -3, but not -4, recruited E3 ubiquitin ligase Mdm2 to the cytoskeleton, with arrestin-1 and -3 increasing the level of ubiquitination of microtubule-associated proteins [65]. Thus, simultaneous arrestin binding to microtubules and other interaction partners appears to localize them to the cytoskeleton and direct Mdm2 activity to its substrates in that compartment. This is reminiscent of the effects of simultaneous arrestin binding to GPCRs and ERK [184].

The localization of GPCR and microtubule binding sites on the same surface suggests that arrestins cannot interact with a GPCR and microtubules simultaneously, i.e., that these are competing interactions. Micromolar affinity of all arrestins for microtubules [65,198], as compared to nanomolar affinity for active phosphorylated GPCRs [46,200], indicates that whenever an active phosphoreceptor is available, it easily wins this competition. As both GPCRs and microtubules engage the same side of the arrestin molecule, the partners that bind to the other side can potentially be recruited by arrestins either to the plasma membrane or endosomes where the receptors reside, or to the cytoskeleton. Microtubule binding induces a specific conformation of arrestin that differs from both basal and GPCR-bound [198]. Conceivably, there might be arrestin-binding partners that prefer that particular conformation, although none has been described so far. Considering that in contrast to ERK1/2, its upstream kinases Raf1 and MEK1 are not recruited to the cytoskeleton [65], these partners apparently “dislike” the conformation of microtubule-bound arrestin.

### 5.2. MAP Kinases

Three-tiered mitogen-activated protein kinase (MAPK) cascades are conserved in all eukaryotes, from yeasts to mammals [201]. These cascades consist of upstream-most MAP kinase kinase kinases (MAP3Ks), intermediate MAP kinase kinases (MAP2Ks), and effector MAPKs [201], which sequentially activate downstream kinase by phosphorylation. Humans express 11 MAPKs, 7 MAP2Ks, and at least 20 MAP3Ks [202]. MAPKs often have multiple splice variants; e.g., JNK family kinases are encoded by three genes, but there are at least ten different JNK isoforms: four of JNK1, four of JNK2, and two of JNK3 [203]. Successful signal transduction requires a combination of matching MAP3K, MAP2K, and MAPK. Because of high protein phosphatase activity in the cytoplasm, if the components of the cascade freely diffused and encountered each other only by chance, an upstream kinase would likely get deactivated by dephosphorylation before encountering the next kinase in line. Therefore, productive MAPK activation cascades are assembled by scaffold proteins, which ensure the correct combination of the kinases and bring them into close proximity for efficient signaling. Importantly, the dependence of MAPK activation on scaffold concentration is biphasic (bell-shaped) [204]. The reason for this is that at low concentration, scaffold that binds all kinases in the cascade increases the probability of the formation of functional three-kinase complexes, whereas when the concentration of the scaffold exceeds that of the kinases, mostly incomplete and unproductive complexes are formed [204]. Indeed, a scaffold for the JNK activation cascade, JIP1, was first discovered as a suppressor of JNK signaling because its overexpression was used [205]. The three most studied MAPK families are ERKs, JNKs, and p38. In simplistic terms, the kinases of the first family are pro-survival, pro-proliferative, whereas the other two are anti-proliferative, pro-differentiation, and sometimes pro-apoptotic.

Arrestin-3 was found to serve as a scaffold of the ASK1–MKK4/7–JNK3 activation cascade in 2000 [185]. This was reported to be a specific function of arrestin-3, which arrestin-2 cannot perform [185,206]. While the original study clearly documented arrestin-3 interactions only with ASK1 and JNK3, arrestin-3 was later shown to bind MKK4 and MKK7 as well [207,208], indicating that it is a true scaffold of the MAPK activation cascade. The first study reported that JNK3 activation via arrestin-3 requires GPCR stimulation [185], but soon thereafter, this was shown to be a receptor-independent process [206]. An arrestin-3 hinge-deletion mutant that does not bind GPCRs was shown to be an effective facilitator of JNK3 activation [144,190,209]. Moreover, short arrestin-3-derived peptides lacking virtually all GPCR-binding elements were shown to facilitate JNK3 activation in cells [210,211]. Recent functional comparison of a large set of arrestin-3 mutants showed that the ability of arrestin-3 to bind GPCRs and to facilitate JNK3 activation has different, virtually opposite, structural requirements [128].

In 2001, both arrestin-2 and arrestin-3 were shown to scaffold Raf1–MEK1–ERK1/2 activation cascades in response to receptor activation [184]. In the first report, only arrestin interactions with Raf1 and ERK1/2 were documented [184], but subsequent studies demonstrated arrestin interactions with MEK1 [144,212]. The dependence of ERK1/2 activation on the functional state of the receptor was independently confirmed many times [190,213,214,215]. ERK phosphorylation upon GPCR activation is often used as a hallmark of arrestin-mediated signaling. However, ERKs can be activated by GPCRs via various pathways, including arrestin-independent ones [216,217]. Moreover, ERK activation in response to several GPCRs was shown to require G protein activity [218,219], although these findings do not exclude the role of arrestin scaffolding in the process [220,221,222]. Recent findings that arrestin-2 bound to hyper-phosphorylated C-terminal peptide of vasopressin V2 receptor allosterically activates Raf [223] and ERK [224] suggests that in the case of the Raf1–MEK1–ERK1/2 cascade, receptor-bound arrestins might play a more active role than that of simple scaffolds.

Arrestins were also found to scaffold p38-activating cascade, at least in response to kappa-opioid receptor activation [225]. Although many MAP3Ks signal via both JNKs and p38, the upstream kinases participating in this process remain to be identified.

As far as MAPKs are concerned, it is important to keep in mind that they are ultimately regulated by the activity of relevant MAP3Ks. GPCR-bound, as well as free, arrestins function as scaffolds, i.e., they facilitate signaling, but do not initiate it. However, reported activation of Raf and ERK by receptor phosphopeptide complex with arrestin [223,224] might be the beginning of a new twist in this story.

### 5.3. Src Family Kinases

GPCRs activate Src family kinases by both direct interactions and through intermediary proteins [226]. Src was the first non-receptor signaling protein shown to bind receptor-associated arrestins [183]. Subsequently, other members of the Src family were also shown to bind arrestins: Hck and Fgr [227], Yes [228], Fyn [229], and Lyn [230], suggesting that arrestin binding is common among Src family kinases. Of course, this does not mean that every member of the family binds arrestins. The interactions are likely multi-site: arrestins were shown to engage the catalytic domain of Src [231], as well as the SH3 domain (binds polyproline motifs) of Src (arrestin-2) [232] and Fgr (arrestin-3) [233]. Multi-domain Src family kinases in their basal (inactive) state exist in closed auto-inhibited conformation, stabilized by intramolecular interactions of regulatory SH2 and SH3 domains [234]. Unwinding of this closed structure, by the interactions of regulatory domains with other proteins, is believed to activate these kinases [234]. Indeed, arrestin interaction with the SH3 domain of the kinase results in the activation of Src [232] and Fgr [233], suggesting that arrestins employ the same mechanism as other activators. It should be noted, however, that hybrid GPCRs were used to bind purified arrestin-2 in the Src study [232] (M2 muscarinic and β2AR with the C-terminus replaced with a multi-phosphorylated C-terminus of the V2 vasopressin receptor, which was synthesized and added to the receptors by sortase) and that no GPCR effect on arrestin-3-dependent activation of Fgr in the cell was detected [233].

## 6. Ubiquitin Ligases and Deubiquitinating Enzymes

Arrestin-3 interaction with E3 ubiquitin ligase Mdm2 was discovered in 2001 [181]. The original study showed that the ubiquitination of both arrestin-3 and activated β2AR was necessary for receptor internalization, so the first hypothesis was that Mdm2 ubiquitinates both proteins. Subsequent study demonstrated that receptor is ubiquitinated by a different arrestin-binding E3 ubiquitin ligase, Nedd4 [235]. Arrestin-dependent ubiquitination of insulin-like growth factor-1 receptor, which is not a GPCR, was later shown to be required for the down-regulation of this receptor [236], as well as for ERK activation and cell cycle progression upon its stimulation [237]. Arrestin-3 was shown to bind the ubiquitin-specific protease USP33 that deubiquitinates β2AR [238]. Arrestin-2 was also shown to interact with another E3 ubiquitin ligase, atropine-interacting protein 4 (AIP4) [239], and both non-visual subtypes, as well as their separated domains, were shown to bind yet another E3 ubiquitin ligase, parkin [146]. Interestingly, parkin binding enhanced arrestin interactions with Mdm2 [146]. As experiments with hinge-deletion mutants incapable of GPCR binding showed, both Mdm2 and parkin prefer the basal conformation of non-visual arrestins [146], which puts the GPCR dependence of these interactions in doubt. Arrestin-3 binding to parkin was shown to enhance parkin enzymatic activity [240]. Moreover, arrestin-3 was shown to rescue the activity of enzymatically impaired disease-causing parkin mutant R275W [240], suggesting that arrestin-3-based molecular tools might prevent the root cause of Parkinson’s disease, namely the demise of dopaminergic neurons in substantia nigra.

## 7. Calmodulin, NSF, and Other Proteins

Direct binding of all four arrestin subtypes to calmodulin was discovered in 2006 [189]. The interaction of the two non-visual arrestins with calmodulin was later independently confirmed by a different lab [75]. Only calcium-liganded calmodulin binds arrestins, and the deletion of the arrestin-2 C-terminus does not preclude this interaction [189]. Calmodulin-dependent reduction in the mobility of spin labels on the surface of arrestin-2 suggests that Ca^2+^-calmodulin binds to the same side of the molecule as GPCRs and microtubules [189], indicating that only free arrestins can bind Ca^2+^-calmodulin. Arrestin-2 affinity for Ca^2+^-calmodulin (K_D_ ~ 7 μM) is higher than for microtubules (see Section 5.1), but orders of magnitude lower than for the active phosphorylated GPCRs, so that in competition, the receptors win. Elevated calcium was shown to slow down intracellular diffusion of ~17 kDa calmodulin, which was interpreted as the result of Ca^2+^-dependent binding of calmodulin to other proteins [241]. It is possible that ~45 kDa arrestins are among these proteins. Arrestins are predominantly cytoplasmic [69,242], and their shuttling between the nucleus and cytoplasm is controlled by the nuclear pore machinery [69,242], whereas calmodulin, being a small protein, freely diffuses between these compartments. Most cells express a lot more calmodulin than both non-visual arrestins put together, so it does not seem likely that arrestin binding can significantly change subcellular localization of calmodulin. The situation is opposite in rod photoreceptors expressing very high levels of arrestin-1 [66], but most of arrestin-1 in rods exists in oligomeric form, where the elements involved in calmodulin binding are shielded by sister protomers [61]. If the affinity of arrestin-1 for Ca^2+^-calmodulin is similar to that of arrestin-2, which has been determined [189], it is higher than its affinity for itself, suggesting that at high calcium, the binding of Ca^2+^-calmodulin has the potential to increase the concentration of monomeric arrestin-1. However, so far no functional consequences of the binding of any arrestin subtype to Ca^2+^-calmodulin have been elucidated.

Arrestin-2 was shown to bind NSF (N-ethylmaleimide-sensitive factor) as early as 1999 [243], but functional effects of this interaction remain unknown. Later, arrestin-1 was also shown to bind NSF, increasing its ATPase activity [192]. NSF plays an important role in the disassembly of the SNARE complex in neurons, enabling the recycling of vesicle membrane and proteins after neurotransmitter secretion. The absence of arrestin-1 in rods precludes the secretion of glutamate, a neurotransmitter used by photoreceptors to signal to bipolar cells [192].

Both non-visual arrestins were shown to interact with more than 100 different proteins each, both upon activation of β2AR and in unstimulated cells [75]. Functional consequences of the majority of these interactions still remain to be elucidated.

## 8. Different Signaling Proteins Preferentially Bind Arrestins in All Known Conformations

Arrestin-dependent activation of the protein kinases c-Src [183], JNK3 [185], ERK1/2 [184], and p38 [225] was first reported to be triggered by GPCR activation, suggesting that these are functions of receptor-bound arrestins. Subsequent studies showed that while this is correct in some cases, it is not a universal rule. Every study performed to date supports the dependence of arrestin-assisted ERK1/2 activation on the formation of the arrestin–GPCR complex. While ERK1/2 bind with much higher affinity to receptor-associated arrestins, their upstream activator kinases MEK1 and Raf1 do not show a clear preference for receptor-bound arrestins [213]. It has been experimentally established that the ability of arrestin-3 (but not other isoforms, even highly homologous arrestin-2 [43,48]) to facilitate the activation of JNK family kinases does not depend on receptor binding: it is a function of free arrestin-3, likely in its basal conformation [144,206,207,208,209,244]. Simultaneous analysis of the activation of ERK and JNK in the same cell showed that ERK phosphorylation was strictly dependent on the activation state of the receptor, whereas the JNK phosphorylation was not affected by the receptor state at all [190]. As far as interactions are concerned, there is no general rule: some arrestin-binding proteins prefer receptor-bound conformation, like ERK1/2 [213], clathrin, and AP2 [170]; some prefer a basal state, like Mdm2 [69,191] or parkin [146]; whereas others appear to bind both, like JNK3, MEK1, c-Raf1, etc. [69,144,191,213]. The preference is not absolute: the partners that prefer the receptor-bound state, such as ERK and clathrin, also interact with free arrestins [168,173].

The biological significance of conformational preferences of arrestin-binding partners is obvious in some cases, and remains to be elucidated in others [182]. For example, preferential binding of key components of the internalization machinery clathrin and AP2 to the receptor-bound arrestins ensures that arrestins promote GPCR endocytosis, and free arrestins in the cytoplasm do not inhibit this process [182]. This is ensured by the anchoring of the C-terminus, which carries clathrin- and AP2-binding sites, to the body of the arrestin molecule (Figure 1) and its release upon receptor binding [134,137,138,142,245]. Apparently, preferential interaction of ERK1/2 with receptor-bound arrestins [213] ensures the role of GPCR in controlling ERK activity, but the biological significance of this phenomenon remains unclear: the activation of all MAPKs, including ERK1/2, is regulated by the activity of upstream-most MAP3Ks, with arrestins acting as scaffolds for the ERK1/2 activation cascade that do not initiate the signaling, only facilitate its propagation. Mdm2 preference for free arrestins [69], in conjunction with the finding that Mdm2 ubiquitinates arrestins recruited to the GPCRs [146,181,238], likely results in only limited ubiquitination, as bound Mdm2 is released from receptor-associated arrestins.

Functional significance of comparable binding of some signaling proteins, such as JNK3 [69], MEK1, c-Raf1, MKK4, and MKK7 [144,207,208,213], to free and receptor-bound arrestins is also unclear. It was shown that JNK3 and the upstream kinases MKK4, MKK7, and ASK1 bind a small, 25-residue peptide of arrestin-3 [210], and even smaller 16-residue part of this peptide [211], which is not an element of any arrestin switch regions [246] and is unlikely to have a preferred conformation as a separate entity. In general, an enormous number of proteins were reported to bind arrestins [75]. The conformational preferences of very few arrestin-binding proteins have been elucidated. We know even less about the biological significance of these preferences. Several signaling proteins also bind microtubule-associated arrestins [65,213], but the role of these interactions also remains obscure.

While the identity of arrestin conformations and the ability of arrestins to change shape were discovered in recent years [247], comprehensive elucidation of the biological role of these changes requires further investigation. Available data suggest that distinct arrestin conformations promote different branches of the arrestin-dependent signaling [248,249]. However, existing data are too far from being comprehensive enough to draw general conclusions. It makes perfect biological sense that different patterns of GPCR phosphorylation by distinct GRKs are decoded by arrestins that assume a different conformation upon binding (barcode hypothesis [155,250,251]). While there is some structural evidence supporting this notion [94,252], the jury is still out.

To obtain unambiguous answers, we need co-structures of arrestins with differentially phosphorylated GPCRs and various downstream signaling proteins. However, the complexes of three or more proteins including GPCRs, which are integral membrane proteins, are unlikely to become available for structural analysis any time soon. As several arrestin-receptor structures have already been solved [79,87,88,89,90,91,92,93], the structures of the same non-visual arrestin with the same GPCR phosphorylated at different positions would be illuminating. Hopefully, these structures would reveal conformational differences in bound arrestin induced by distinct patterns of receptor phosphorylation. The structures of arrestin complexes with non-GPCR signaling proteins might reveal subtle conformational preferences of different binding partners. Collectively, these structures will improve our understanding of the regulation of arrestin-dependent signaling and also hopefully support the barcode hypothesis, which appears very attractive because it makes perfect sense biologically.

## 9. Arrestin-Assisted Signaling: GPCR-Dependent and Independent

Arrestins were shown to regulate numerous signaling pathways in the cell. Structural studies demonstrated that the conformations of free and receptor-bound arrestins are quite different. The functions of these two states of the arrestin molecule are also different, some specific for the receptor-bound form, others performed exclusively by free arrestins. It is likely that there are functions that arrestins can perform in both conformations.

Only in three cases was arrestin-mediated signaling unambiguously shown to require arrestin binding to a GPCR: the activation of Src [183,232], ERK1/2 [190,253], and focal adhesion kinase via STAM1 (signal transducing adaptor molecule 1) [254]. The list of arrestin functions that were demonstrated not to require GPCR binding is longer: caspase cleavage and regulation of apoptosis by caspase-generated fragments [255,256], regulation of cell spreading and motility by disassembling focal adhesions [173] and activation of small GTPases RhoA and Rac1 [257], activation of NSF [192], activation of enolase-1 [193], and regulation of the activity of protein kinases MELK (maternal embryonic leucine-rich kinase) [258], Fgr [233], and JNKs [128,144,190,207,244].

Existing data suggest that some arrestin-binding partners interact with receptor-bound arrestins with the highest affinity, some with free (basal) forms, some with microtubule-associated forms, whereas there are partners that do not show a clear preference, apparently engaging arrestin elements that are exposed in all of these states and do not change much. Although conformational preferences of most arrestin-binding partners remain unknown, available data show that there is no such thing as an inactive arrestin conformation: every conformation is active in certain signaling pathways and inactive in others.

## 10. Arrestin-Based Molecular Tools

Because arrestins regulate many vital signaling pathways in the cell, they appear to be a useful “handle” to manipulate these pathways for research and potentially therapeutic purposes. Two types of arrestin-based molecular tools have been described so far: engineered signaling-biased mutants and monofunctional peptides extracted from multi-functional arrestin proteins.

### 10.1. Signaling-Biased Full-Length Arrestins

Point mutation V343T in bovine arrestin-3 greatly reduces its ability to facilitate the activation of JNK3 in cells [209]. As this mutation is on the non-receptor-binding side, it is unlikely that GPCR binding was affected, but this and other functions were not tested. Unexpectedly, it was found that arrestin-3–KNC mutant that does not bind GPCRs because ten key receptor-binding residues in it were replaced by alanines fails to facilitate the activation of JNK3, whereas another non-receptor-binding mutant with seven residue deletions in the interdomain hinge is efficient in this regard [128,190]. Arrestin-3–KNC binds JNK3, and therefore acts as a dominant-negative “silent scaffold”: it recruits JNK3 away from productive scaffolds (including wild-type arrestin-3), thereby suppressing its activation [190]. Thus, arrestin-3–KNC has two known functional peculiarities: it does not bind GPCRs [128] and it does not activate JNK3 [190]. Its other functional capabilities have not been tested. Arrestin-2 point mutant R307A demonstrates greatly reduced binding to Raf1, the MAP3K of the Raf1–MEK1–ERK1/2 cascade [253]. Even though this mutant binds GPCRs, MEK1, and ERK1/2 normally, it does not facilitate ERK1/2 activation in cells in contrast to the parental arrestin-2 [253]. Interestingly, in arrestin-3, this position is occupied by a lysine (ostensibly a conservative substitution), but the side chain of the homologous K308 in arrestin-3 points in a different direction in the structure [253]. Other functions of arrestin-2-R307A were not tested.

Several mutants of arrestin-1 [82,98,107,111,259], as well as arrestin-2 and -3 [99,100,114,127,128], with either the polar core or the three-element interaction (Figure 1) destabilized, were shown to bind phosphorylated and unphosphorylated cognate GPCRs much better than parental wild-type proteins. Two structurally different phosphorylation-independent arrestin-2 mutants were shown to accelerate the cycling of β2AR and prevent its down-regulation upon prolonged stimulation in cells [114]. Arrestin-3 mutants with enhanced GPCR binding were shown to be ineffective in JNK3 activation [128,190].

So far, only three engineered mutants with special functional characteristics, all of visual arrestin-1, have been tested in vivo. One was a variant of mouse arrestin-1 (arrestin-1-3A) that binds unphosphorylated Rh* much better than the wild-type protein (the C-terminus of this mutant is pre-released by triple alanine substitution that prevents its participation in the three-element interaction (Figure 1)) [108]. This mutant was shown to partially compensate for the lack of rhodopsin phosphorylation [260,261]. The second was oligomerization-deficient mouse arrestin-1 double mutant F86A+F198A [59]. This non-oligomerizing arrestin-1 fully substitutes for the wild-type protein in quenching rhodopsin signaling, but at higher expression levels causes light-independent death of rod photoreceptors [68]. Interestingly, arrestin-1-3A, which also oligomerizes less readily than parental wild-type protein [259], similarly causes photoreceptor death that does not depend on light exposure, with higher expression level being more detrimental [262]. These data suggest that arrestin-1 oligomerization protects rods from harmful effects of high levels of monomeric arrestin-1 [68]. The third was arrestin-1 mutant with increased ability to stimulate the activity of enolase-1. Enolase-1 converts 2-phosphoglycerate into phosphoenolpyruvate, which is then converted to lactate. Lactate is provided by rod photoreceptors to retinal pigment epithelium and Müller glia for their energetic needs. Arrestin-1 mutant that stimulates enolase-1 more effectively than the wild-type protein increases lactate output of photoreceptors in vivo and slows down the decline in the photoreceptor function and retinal degeneration in mice with disease-causing rhodopsin mutation, suggesting that it might be suitable for gene therapy of retinal degeneration [193].

### 10.2. Monofunctional Arrestin Elements

Arrestins are remarkably multifunctional proteins [74]. As the internal workings of the arrestin molecule are poorly understood, targeted manipulation of one function often results in unintended changes in others [128,190,262]. Because of pleiotropic effects of mutations, signaling-biased full-length arrestins, while useful as research tools, are unlikely to become therapeutically usable. Therefore, extracted monofunctional arrestin elements that can do only one thing and nothing else hold a greater promise as molecular tools with therapeutic potential. Two of these have been constructed and tested: the C-terminus of arrestin-2 harboring clathrin- and AP2-binding sites, and arrestin-3-derived peptides that scaffold the ASK1–MKK4/7–JNK3 signaling module.

The expression of separated C-terminus of arrestin-2 inhibited GPCR internalization, apparently via competition with the arrestin-receptor complexes for clathrin and AP2 in cells [263]. This peptide has no GPCR-binding elements and is not known to interact with other arrestin-binding partners. It might prevent excessive internalization and consequent down-regulation of GPCRs, e.g., that of β2AR, which appears to underlie congestive heart failure [264]. However, by the mechanism of action, it is likely to simultaneously suppress the endocytosis of all GPCRs that internalize via coated pits in an arrestin-dependent manner. Thus, its effects in cells are likely to be pleiotropic.

Several fairly short peptides from the arrestin-3 sequence were shown to facilitate JNK3 activation both in vitro and in cells [210,211]. Homologous arrestin-2-derived peptides lack this activity [210]. There are virtually no GPCR-binding elements in these peptides, and their length (14–25 residues out 408 in arrestin-3) makes it unlikely that they interact with other proteins that arrestin-3 binds. As activators of anti-proliferative JNK signaling, they have potential in disorders associated with excessive cell proliferation, such as cancer. These peptides appear to bind all kinases in the ASK1–MKK4/7–JNK3 signaling module [211]. On the basis of the mechanism of the scaffold function, engineered modified and/or shortened peptides that do not assemble a complete productive signaling module are likely to suppress JNK signaling in dominant-negative fashion, like the arrestin-3–KNC mutant [190]. Tools of this type might be beneficial in degenerative disorders, such as Alzheimer’s or Parkinson’s diseases, where excessive JNK3 activity appears to result in neuronal death.

## 11. Conclusions

Arrestins are average-sized multifunctional signaling regulators, with binding to active phosphorylated GPCRs being only one of their biological functions. Arrestins have no enzymatic activity; they only function by interacting with other proteins, assembling multi-protein complexes, and delivering their binding partners to particular intracellular compartments. Arrestins affect numerous signaling pathways in the cell, including those that determine cell fate. Therefore, arrestins appear to be a tempting handle for targeted manipulation of cell signaling for research and therapy. However, multifunctionality of full-length arrestins suggests that signaling-biased mutants might be useful for research, but would produce too many unwanted side effects as therapeutic tools. Small monofunctional peptides extracted from arrestin proteins can regulate certain branches of signaling. Due to low probability of pleiotropic effects, arrestin-derived peptides appear to be molecular tools with greater therapeutic promise.

## Figures and Tables

**Figure 1 ijms-25-06284-f001:**
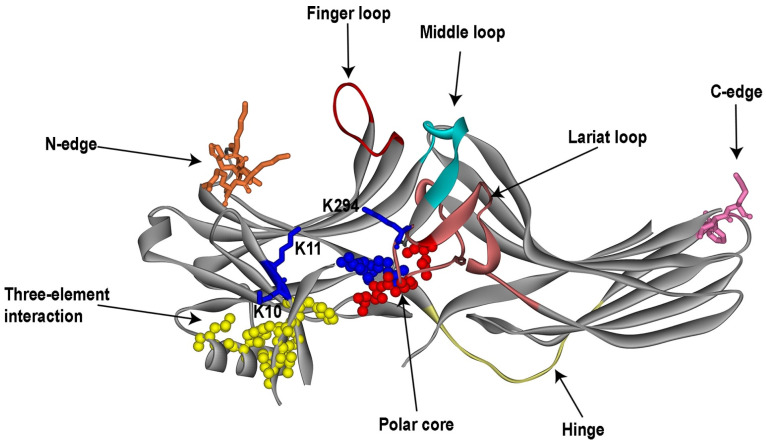
**Functional elements of arrestins.** Crystal structure of bovine arrestin-2 (PDB ID 1G4M [49]). Functional elements are indicated as follows: residues in the polar core (Asp26, Arg169, Asp290, Asp297; Arg 393 in the C-terminus is not resolved in this monomer and therefore is not shown) and three-element interaction (Val8 and Phe9 in β-strand I; Leu100, Leu104, and Leu108 in the α-helix; and Ile386, Val387, and Phe388 in the C-terminus are not resolved in the structure of this monomer and therefore are not shown) are shown as CPK models, and Lys10, Lys11, and Lys294, as well as C-edge residues 190–192 and N-edge residues 157–161 are shown as stick models. The following elements are highlighted by color: the finger loop (residues 64–74; red), the middle loop (residues 129–139; light blue), the lariat loop (residues 275–316; light red), N-edge (residues 157–161; orange), C-edge (residues 190–192; pink; note that the other C-edge loop, residues 334–338, is not shown because it is not resolved in the structure of this monomer), and inter-domain hinge region (residues 173–184; yellow). The chemical nature of the modeled residues is shown, as follows: hydrophobic, yellow; positively charged, dark blue; negatively charged, bright red. Image was created in DS ViewerPro 6.0 (Dassault Systèmes, San Diego, CA, USA).

**Figure 2 ijms-25-06284-f002:**
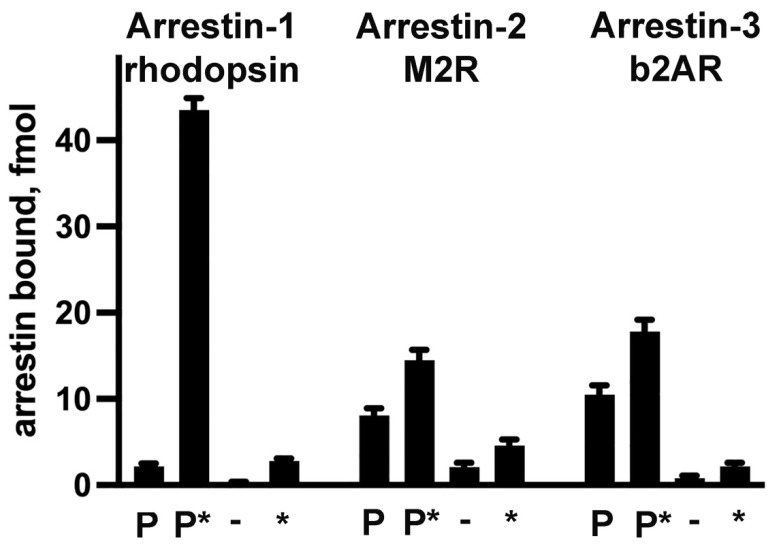
**Binding selectivity of arrestins.** The binding of arrestin-1 to rhodopsin, arrestin-2 to M2 muscarinic receptor (M2R), and arrestin-3 to β2-adrenergic receptor (b2AR) is shown. Functional forms of the receptors are indicated, as follows: P, inactive phosphorylated; P*, activated phosphorylated; -, inactive unphosphorylated; *, activated unphosphorylated.

**Figure 3 ijms-25-06284-f003:**
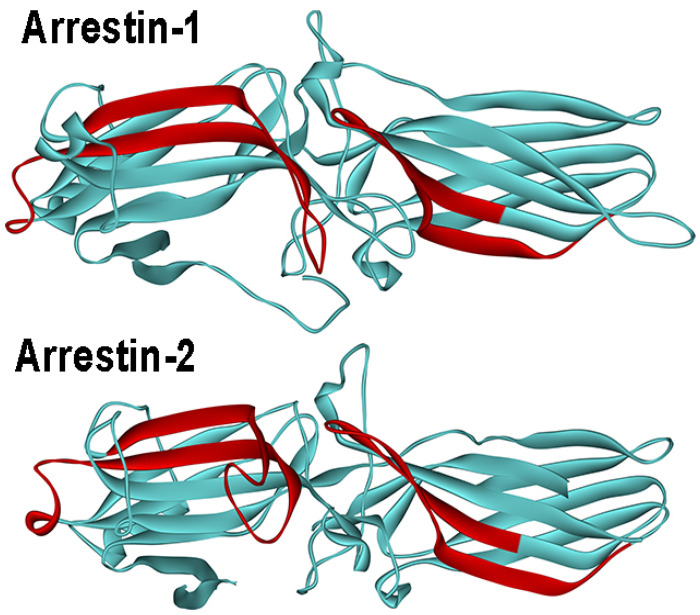
**Arrestin elements that determine receptor preference.** These were identified in [105]: arrestin-1 residues 49–90 in the N-domain and 237–268 in the C-domain, and homologous arrestin-2 residues 45–86 and 233–262. The structures are shown as solid ribbons. Receptor-discriminator elements are shown in red on the structure of arrestin-1 (PDB ID 1CF1 [20]) and arrestin-2 (PDB ID 1G4M [49]). The remaining arrestin-1 and -2 molecules are shown in blue. Images were created in DS ViewerPro 6.0 (Dassault Systèmes, San Diego, CA, USA).

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
