# Peer review of "Arrestins: A Small Family of Multi-Functional Proteins"

_ijms, 2024, doi:10.3390/ijms25116284_

Round 1
Reviewer 1 Report
Comments and Suggestions for Authors
This review covers structural and biochemical aspect of arrestins. After a historical introduction, arrestin structure/modes of self-association are described and GPCR binding is discussed at length before a short paragraph on the role of arrestin in GPCR trafficking. Arrestin interactions with a few non-receptor binding partners, with ubiquitin/deubiquitinating enzymes as well as with calmodulin and NSF are next described. Finally, the roles of arrestin specific conformations in their interaction with protein partners, the question of arrestin dependence on GPCR for signaling and the “molecular tools” useful to study arrestin signaling are discussed.
It is a very nice and comprehensive review. Some sections nicely recapitulate and put together recent more detailed reviews on specific characteristics/properties of arrestins by this author.
The author often chose to organize the manuscript on the basis of structural studies mostly performed on arrestin 1 by his laboratory and to extend the discussion to other arrestins. It could have been organized differently to better reflect the chronology of certain innovative findings on arrestin 2 and 3 but it is a fair choice.
I only have one main remark. The manuscript is organized on the basis of the research interest and results of the author's laboratory. As such, more than 40% of the cited references are from the author of the review. Naturally, the citations, which are appropriate, also reflect the impressive number of published papers (research and reviews) by the author on these questions but the contribution of other laboratories to the field could also have been more cited.
My recommendation would be:
- re-equilibrate the ratio between self- and other citations by adding more contributions to the field from other investigators. For example, most citations are self-citations in section 5 and 8. Other work related to the discussed signaling partners of arrestins could be mentioned as well as other signaling partners characterized by other investigators. Also, a very recent paper from Cell Reports describes “ArreSTick” sequences…… Despite its interest, paragraph 6.1 could also be removed…..
- Alternatively, a sentence at the start of the manuscript mentioning that “some important issues related to arrestin biology studied in our lab and others will be discussed” could be added.
Minor comments
- Title of 3.1 and 9.1 are at the bottom of the pages
- “between” lane 496 is mentioned twice
- “arestin” lane 588
- Errors in titles and numerotation of paragraph 6. and 6.1?
- Redundancy between intro section 5 and section 8
Author Response
My recommendation would be:
- Alternatively, a sentence at the start of the manuscript mentioning that “some important issues related to arrestin biology studied in our lab and others will be discussed” could be added.
Thanks! The manuscript focuses on the discussion of the structural basis of known arrestin functions. Unfortunately, most labs studying arrestins limit themselves to functional phenomena (all referenced) and do not dive into structural basis of the functions performed by these proteins. Therefore, although we added several relevant references from other labs, we chose to follow this suggestion rather than re-writing the whole manuscript.
Minor comments
- Title of 3.1 and 9.1 are at the bottom of the pages
Thanks. Reformatted.
- “between” lane 496 is mentioned twice
Thanks for your attention to detail! Corrected.
- “arestin” lane 588
Thanks! Corrected.
- Errors in titles and numerotation of paragraph 6. and 6.1?
Thanks! Corrected. Please also note that section 6.1 was included in section 6 by mistake. Now it is a separate section.
- Redundancy between intro section 5 and section 8
Thanks! The introductory paragraph in section 9 (former section 8) was re-written to avoid redundancy.
Reviewer 2 Report
Comments and Suggestions for Authors
This review describing and discussing the multifunctional role of arrestins in GPCR biology (and beyond) is very well written and clearly structured. It contains a lot of information that is critically discussed. This is a very useful review for the GPCR community.
Few minor typos:
Figure 1: Middle "loop" instead of Middle "llop".
line 110: color instead of clolor.
line 150: correct "oligopmeric"
line 183: negatively instead of negitively
Figure 2: correct Y-axis ("arrestin bound, fmol)
Line 588: arrestin instead of arestin
Author Response
Few minor typos:
Figure 1: Middle "loop" instead of Middle "llop".
line 110: color instead of clolor.
line 150: correct "oligopmeric"
line 183: negatively instead of negitively
Figure 2: correct Y-axis ("arrestin bound, fmol)
Line 588: arrestin instead of arestin
Thank you for careful reading of this lengthy manuscript! All indicated errors were corrected.